# The Study of Radius End Mills with TiB_2_ Coating When Milling a Nickel Alloy

**DOI:** 10.3390/ma16062535

**Published:** 2023-03-22

**Authors:** Sergey Grigoriev, Marina Volosova, Mikhail Mosyanov, Sergey Fedorov

**Affiliations:** Department of High-Efficiency Processing Technologies, Moscow State University of Technology STANKIN, 127055 Moscow, Russia

**Keywords:** ball-end mills, coatings, DLC, TiB_2_, abrasive and adhesive wear

## Abstract

Nickel alloy high-speed processing technology using ball-end mills is characterized by high contact temperature and leads to accelerated tool wear. One of the effective ways to increase its reliability and service life is to modify the surface by applying functional antifriction layers in addition to wear-resistant coatings. Diamond-like carbon is often used as the latter. However, at cutting speed, when a cutting-edge temperature exceeding 650 °C is reached, the material of this coating reacts actively with oxygen in the air, and the sharply increasing adhesive component of wear quickly incapacitates the milling tooth, limiting its performance. Applying a coating of titanium diboride as an antifriction layer on top of nanocrystalline composite nitride coatings with good resistance to abrasive wear can be a solution to this problem. Our experiments have shown that such technology makes it possible to obtain a twofold increase in durability compared to a tool with a diamond-like antifriction coating in conditions when the cutting edge of the tool is subjected to cyclic thermal shocks above 800 °C, and the durability period of the radius end mill is about 50 min.

## 1. Introduction

The processing of heat-resistant nickel-based alloys accounts for a significant share in the classification of processed materials, which belong to the class of hard-to-process materials. When milling labor-intensive products from them, significant difficulties arise due to the low resource of the tool. At the same time, the processing complexity only increases, which explains the development of new heat-resistant alloys to meet modern requirements, and it is necessary to ensure acceptable conditions by reducing cutting modes, and this, in turn, leads to a decrease in productivity, chips on the cutting edge, and increased wear.

The main areas of research in cutting tool technology for processing nickel alloys are related to the creation of new tool materials, tool geometry, and improvement of processing methods [1]. The high-speed processing method of elaboration of complex shape surfaces is associated with expanding the use of ball-end mills. The advantage of this technology is a noticeable reduction in surface roughness with proper diagnostics of the intermittent cutting process [2], not to mention the possibility of excluding some operations. However, along with the obvious advantages, this technology poses several issues for handlers associated with increased cutting temperature, causing accelerated wear of the milling cutter. In addition, the lack of rigidity in the system leads to the appearance of radial runout and, consequently, uneven load on the cutting teeth [3].

Recognized leaders in cutting tool production are developing all-ceramic mills for various purposes [4]. However, working with such tools requires appropriately sophisticated technological equipment. Therefore, the current most widely used tool in Ni-based superalloy manufacturing is still a carbide tool with an average grain size of sintered carbide of less than 1 µm due to the excellent balance between cost and performance, especially if the cutters are equipped with a modern wear-resistant coating.

Another promising direction is related to the texturing of functional surfaces, which improves cutting characteristics, ensuring the processing quality and affecting the wear degree due to changes in lubrication conditions. Tool durability increases due to reduced cutting forces and temperature, friction forces, and chemical activity of materials in the contact zone [5]. Although additional operations in the manufacture of the tool slightly increase its cost, the advantages of textured cutters and plates become evident in the cost of the workpiece [6].

One of the promising ways to improve the characteristics of a carbide tool is to modify the working surfaces by saturation with alloying elements. A modified layer is created on the surface using electron beam alloying technology due to the initiation of exothermic chemical reactions between the carbide substrate and the thin film deposited on it. The use of such technology and the application of a wear-resistant coating can significantly increase the tool’s durability, and the effect is most noticeable in intensive cutting modes when the stresses arising in the cutting-edge zone reach values close to the strength limit of a hard alloy [7,8].

It is also important to mention the standard modern method of increasing the reliability and a cutting tool’s service life, multilayer composite coating application [9,10], as well as the surface modification by applying functional coatings on top [11]. Among the various types of such coatings, diamond-like carbon (DLC) is of specific interest, having a low coefficient of friction, high hardness, and wear resistance. It is suitable for tribological applications, particularly for processing heat-resistant alloys [12,13]. The secondary structures appear on the friction surface of the sample with DLC due to carbon phase saturation with nickel and chromium and their oxides. This phase can influence the evolution of fatigue failures in stress concentration zones. Their beneficial tribological properties are associated with sliding phenomena occurring in the transition layer functioning as a solid lubricant and formed in the friction contact zone due to graphitization and diamond-like coating oxidation [14].

However, studies have shown that the temperature near the cutting edge is a parameter that significantly limits the cutting ability of diamond-like end mills with DLC coating when processing nickel alloys, especially in dry milling conditions [15,16]. It has been determined that the wear-resistant coating based on (CrAlSi)N with applied DLC film is effective for solving the technical problem of increasing the tool cutting ability, as well as for improving the treated surface roughness, but only for conditions when the temperature effect on the surface layer does not exceed 650 °C.

Of interest are studies concerning coatings based on titanium diboride, which have high thermal and chemical stability and a sufficiently low coefficient of friction (about 0.4). A significant self-lubricating effect of TiB_2_ coatings was observed, associated with forming B_2_O_3_ tribe films when interacting with oxygen from the environment [17], which reduces friction at the tool–chip interface. The coating is positioned as a composition for processing non-ferrous metals, mainly aluminum and titanium, which are characterized by the intensive formation of sticking on the tool cutting edge. In most cases, the TiB_2_ coating is obtained by magnetron sputtering, since the cathode-arc evaporation of this material is associated with some technical difficulties [18]. It is usually a monolayer with a thickness of 1 to 4 µm.

The authors have not seen publications describing titanium diboride as an antifriction layer on top of a wear-resistant nitride coating for processing nickel alloys by analogy with standard coatings, where the DLC film is applied to the nitride wear-resistant coating. The purpose of this work was a comparative study of the radius end mills with antifriction coatings TiB_2_ and DLC applied to a composite nanocrystalline coating based on (Ti(Cr)AlSi)N cutting conditions when milling a heat-resistant nickel alloy.

## 2. Materials and Methods

### 2.1. Cutting Tools and Machined Material

The studies were carried out on experimental ball-end mills. Their geometric parameters are controlled by a Helichek Plus machine (Walter, Olpe, Germany), and are shown in Table 1. Figure 1 shows a general view of the ball-end mill design.

The milling cutters were manufactured on the La Prora U320 grinding machine (LTF, Antegnate, Italy) from a calibrated rod of KFM 39 tungsten carbide manufactured by Konrad Micro Drill (Kulmbach, Germany). Table 2 shows the characteristics of the alloy.

A workpiece made of heat-resistant nickel alloy Ni_45_Fe_30_Cr_14_Mo_4_W_3_Ti_2_NbAl was processed (indexes indicate wt.%). The structure of this alloy is an austenitic solid solution of the Ni-Cr-Fe system. In practice, the alloy is used for highly loaded elements of the supporting structure and other parts of gas turbine engines operating in various climatic conditions at temperatures up to 800 °C. The workpiece in this experiment was a hot-rolled rod with a diameter of 55 mm. The alloy had a hardness of 320 HB and a strength of 1080 MPa.

### 2.2. Investigation of the Cutting Tool Properties

The experimental ball-end mills were tested on a CTX beta 1250 TC lathe (DMG MORI (Bielefeld, Germany) equipped with a Siemens CNC system (Munich, Germany)). The cutting scheme is shown in Figure 2. The milling cutter fixed in the collet chuck, mounted at an angle of 40° to the surface of the workpiece, performed the main rotational movement with the cutting speed *V_c_* and the cutting depth *a_p_*, moving along the axis of the workpiece with the feed *f_z_*. In turn, the cylindrical workpiece rotated in a three-cam cartridge at a speed of *V_w_*. The cutting mode that provides the necessary thermal load on the cutting part of the end mill is shown in Table 3.

As a criterion for tool failure, the wear value on the flank surface was selected to equal 0.2 mm, which was measured using the instrumental optical microscope Discovery V12 Carl Zeiss (Oberkochen, Germany). In addition, the wear zone was analyzed using a MicroCAD-lite 3D scanner (GFM, Graz, Austria) and a MIRA3 scanning electron microscope (Tescan, Brno, Czech Republic).

### 2.3. Application of Wear-Resistant Coatings on Experimental Milling Cutters

The wear-resistant coatings presented in Table 4 were applied on Platit π411+ (Platit, Switzerland), which was equipped with a SCIL module (sputtered coating induced by lateral glow discharge) [19] using a double planetary rotation rig when the vacuum chamber was fully loaded. CrN−ncAlTiN/Si_3_N_4_ and CrN−ncAlCrN/Si_3_N_4_ were used as coatings that played the role of a sublayer for the antifriction composition of TiB_2_. They were obtained using the PVD method from cylindrical rotating cathodes Ti, AlSi18%, and Cr. This class of nanocomposite coatings was specially developed to counteract high-temperature wear [20,21]. Their unique properties are provided by the presence in the structure of at least two phases: nanocrystalline, consisting of crystallites (Ti(Cr)AlSi)N with a size of about 5 nm, and an amorphous Si_3_N_4_ matrix [22]. The phase boundaries are an area of intense energy dissipation that deflects cracks in the coating from their trajectories and reduces the speed of their propagation, which leads to the solidification of the material.

The process started with the deposition of a contact layer composed of Cr adhesion and CrN transition layers, followed by a (Ti(Cr)AlSi)N main gradient layer. The coating was formed by the current changing of the Ti(Cr) and AlSi cathodes from 125 to 100 A, and from 110 to 140 A, respectively. The bias voltage was maintained at −45 V. The process temperature of 500 °C was maintained using resistive heaters.

The TiB_2_ coating was applied to the initial and previously nanocomposite-coated milling cutters at a process temperature of 400 °C from a 15-kW central magnetron, a bias voltage of 60 V (pulse voltage source: frequency 30 kHz, duty cycle 85%), and an argon pressure of 0.75 Pa. The SCIL technology makes it possible to take full advantage of the magnetron sputtering and cathode arc methods. In this case, the arc is an external ionization source for the magnetron atomization process of titanium diboride targets, providing an acceptable deposition rate and sufficient adhesion of the coating.

The CrN−ncAlCrN/Si_3_N_4_ + DLC coating was applied by installing Platit π311 + DLC (Platit, Switzerland) in one technological cycle, unlike the above processes. A diamond-like coating layer based on a-C:H:Si was deposited by the PACVD method subjected to discharge destruction from acetylene with a 1% addition of tetramethyl silane. Raman spectroscopy showed that the percentage of sp^3^ (diamond) hybridization is 72%. The thickness of the coating and each of its layers was monitored using a Calowear device (GFM, Graz, Austria), and the condition of the milling cutter surface after coating was analyzed with a Dektak XT stylus profiler (Bruker, Billerica, MA, USA). Microhardness was measured by an microhardness meter (Anton Paar, Graz, Austria) with force transducer MB-10.

## 3. Results and Discussion

### 3.1. Structure and Properties of Coatings Deposed to End Mills

All coatings were uniformly formed on the working surfaces of the end mills and did not radically change the radius of rounding of the cutting edge, which was 13–15 µm. Table 4 shows the grinding image, a profile of the surface, and measurement data of the parameters of the coatings under study. The total thickness of the coatings was 2.5–2.7 µm, and the thickness of the antifriction layer was 0.6–0.7 µm. The coating thickness on the sample with TiB_2_ was 2.5 µm. One feature should be noted here. On ceramic substrates, such as the nitride coatings CrN−ncAlTiN/Si_3_N_4_ and CrN−ncAlCrN/Si_3_N_4_, the growth of the TiB_2_ coating, which was applied in one technological cycle to all cutters at the same time, was more than three times slower than directly on the hard alloy, which can probably be attributed to various electrical–physical properties of the substrate.

The surface roughness on cutters with nitride coating and titanium diboride R_Z_ was 1.6–1.7 µm. On the profile, a certain amount of the droplet phase can be observed with a particle size of up to 5 µm in the area and a height of up to 1 µm. Sputtering from an AlSi cathode is responsible for their largest [23]. Tiny chromium droplets, about 1 µm, can also be seen on the TiB_2_-coated sample. They were splashed on the tool when the surface was activated by metal ions. Magnetron sputtering provided much less roughness (R_Z_ = 1.15 µm, R_A_ = 0.06 µm). The sample with CrN−ncAlCrN/Si_3_N_4_ + DLC coating had the most considerable roughness on the R_Z_ = 2.0 µm scale, but on the R_A_ scale, its value practically did not differ from other vacuum-arc coatings, being in the range of 0.11–0.15 µm. On the surface of this milling cutter, the profilometer revealed spherulites up to 3 µm, characteristic of DLC films obtained by chemical vapor deposition.

The microhardness measured at a load of 10 g for all coatings was in the range of HV_10_ = 34–38.4 GPa. At the same time, the indenter penetrated the coating by approximately 0.6 µm, so some influence of the substrate on the measurement result was undoubtedly present.

### 3.2. Cutting Capacity of Coated End Mills

Figure 3 shows the experimentally obtained dependences of the flank surface wear average value of ball-end mill teeth with different coatings on the nickel alloy processing time under the cutting mode specified in Table 3; in particular, the speed *V_C_* = 250 m/min. Processing at such a speed was characterized by the fact that in the area adjacent to the cutting edge, the tool was heated to a temperature exceeding 800 °C [16].

The nature of wear has a classic character. For uncoated tools (curve 1), the milling time until the critical wear 0.2 mm value on the back surface was about 15 min.

Curves 2 and 3 correspond to a tool with TiB_2_ and CrN−ncAlCrN/Si_3_N_4_ + DLC coatings, respectively. The durability of these cutters is approximately the same and amounted to 20–24 min at the specified cutting mode, increasing by 30–50%. They were also united by the fact that at temperatures above 600 °C they are no longer able to restrain the setting processes; in the case of CrN−ncAlCrN/Si_3_N_4_ + DLC, this was precisely because of the DLC component. The reason may be the more significant predominance of the adhesive wear mechanism over the abrasive one, which both coatings resisted well with a lower thermal load. In [24,25,26], the increasing role of the adhesive component of the friction coefficient was also noted, contributing to the setting of the tool and the workpiece with increasing temperature. The rear angle decreases, the friction force increases, the secondary structures formed cannot withstand thermal shocks and can no longer cope with their tribological function, the temperature in the contact zone continues to rise, and the wear rate increases. Structural changes in coatings are accompanied by a drop in their properties, and they are no longer able to achieve the desired effect. The condition of the cutting edges of tools with TiB_2_ and CrN−ncAlCrN/Si_3_N_4_ + DLC coatings and the distribution of chemical elements over the surface obtained using EDX analysis can be estimated in Figure 4 and Table 5.

There is intense build-up on the rake and flank surfaces of the milling cutter, especially with CrN−ncAlCrN/Si_3_N_4_+DLC (Figure 4c,d, Table 5 light cross-section) coating. WC particles broke off from the matrix and embedded in the growth of the processed material, characteristic of the adhesive type of wear.

The presented SEM images show some differences in the wear of the tool with a TiB_2_ coating. The amount of nickel alloy sticking near the cutting edge was noticeably less in this case. However, at the same time, it can be seen, particularly by the light section, that the abrasive component of wear has worked quite seriously with the flank surface of the cutter. Apparently, at a cutting speed of 250 m/min, the CrN−ncAlCrN/Si_3_N_4_+DLC coating resists abrasive wear better, and TiB_2_ restrains its adhesive component better. Nevertheless, it is impossible to recognize the test results as satisfactory in both cases.

An Interesting effect was obtained by combining the coatings of nitrides (Ti(Cr)AlSi)N with TiB_2_. A practically twofold increase in durability was obtained (curves 4 and 5 in Figure 3). This increased wear resistance is due to adaptation associated with the intensifying of nonequilibrium processes during friction. Some studies have shown the predominant formation of surface films of protective tribological ceramics based on nonequilibrium polyvalent protective oxides of titanium, chromium, and aluminum in the surface layers of nitride coatings [27]. At a contact temperature exceeding 700 C, the oxidation of titanium diboride begins [28,29], and a decrease in adhesive setting can be associated with the formation of tribological films based on boron oxide [17]. In addition, it has been shown that interaction in oxide structures is possible under these conditions. In particular, the doping of TiO_2_ with boron oxide affects its crystallinity [30].

Figure 5 and Table 6 show the corresponding images of the cutting edges worn to the failure criterion of the cutters. In the case of CrN−ncAlTiN/Si_3_N_4_+TiB_2_, there is practically no sticking of the processed material directly on the cutting edge, although the adhesive on the back surface covers the contact zone. In Figure 5a, scratches caused by solid particles of the processed nickel alloy are present. In the light section (Table 6), the step caused by abrasive wear is visible. The wear of the milling cutter tooth is gradual and predictable, which is a sign of the regular operation of the tool.

Observations of the wear mechanisms on the surface of coatings allow us to conclude that the composition of the coating should contain a minimum number of elements similar to the treated metal base to minimize adhesive wear of the coating. As expected, the CrN-ncAlCrN/Si_3_N_4_+TiB2 coating worked somewhat worse than the titanium-containing coating, although the tool showed less wear during the first 20 min of operation with it. Scratches from abrasive interaction are also present here, but it can be seen that they are already smeared with the processed material (Figure 5c). The adhesion of the processed material to the rake surface is much more intense here (Figure 5d, Table 6). The interaction of the tool and the workpiece, in this case, was more robust, probably due to chromium, which was present in the coating and the heat-resistant alloy in sufficient quantities.

## 4. Conclusions

A tool with nanostructured composite coatings based on complex nitrides invariably demonstrates good performance when processing nickel alloys. Some authors associate such increased wear resistance with adaptation processes in thin surface layers associated with forming protective oxide tribological films. By applying additional antifriction layers to nitride coatings, for example, the DLC and TiB_2_ films considered in this work, it becomes possible to optimize the structure of the protective wear-resistant coating on carbide ball mills designed for high-speed cutting, depending on the properties of the material being processed and the intended cutting mode.

However, the best functional properties under predetermined conditions correspond to less versatility when the cutting parameters deviate from the set values associated with increased thermal load on the cutting edge. If we adhere to the hypothesis that the temperature and adhesion of the processed material due to the growth of the adhesive component of wear are closely related, it is possible to conclude that B_2_O_3_, which was formed during the oxidation of titanium diboride and positively affected the functional properties of tribological films based on titanium oxides, chromium, and aluminum due to interaction with them, can reduce the temperature in the contact zone by reducing friction and ensure the operation of the intermediate layer of coatings that resist the abrasive component of wear well. Therefore, by combining nitride coatings (Ti(Cr)AlSi)N with TiB_2_, it was possible to obtain a twofold increase in durability compared to the coating with DLC under conditions of pulsed heating of the milling cutter tooth above 800 °C.

## Figures and Tables

**Figure 1 materials-16-02535-f001:**
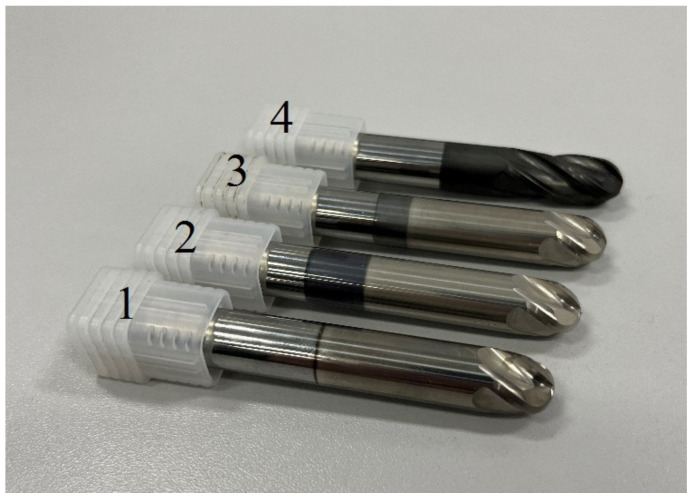
The experimental ball end mills with coatings 1—TiB_2_, 2—CrN−ncAlTiN/Si_3_N_4_ + TiB_2_, 3—CrN−ncAlCrN/Si_3_N_4_ + TiB_2_, 4—CrN−ncAlCrN/Si_3_N_4_ + DLC.

**Figure 2 materials-16-02535-f002:**
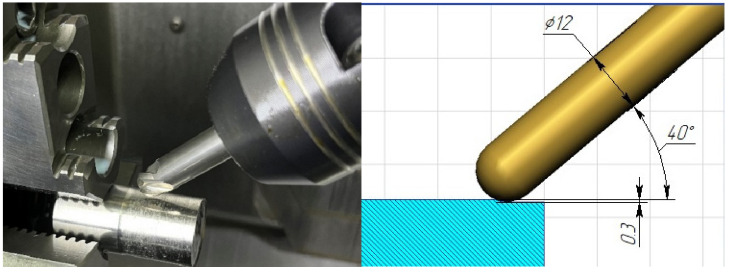
The milling–turning test on ball-end mills.

**Figure 3 materials-16-02535-f003:**
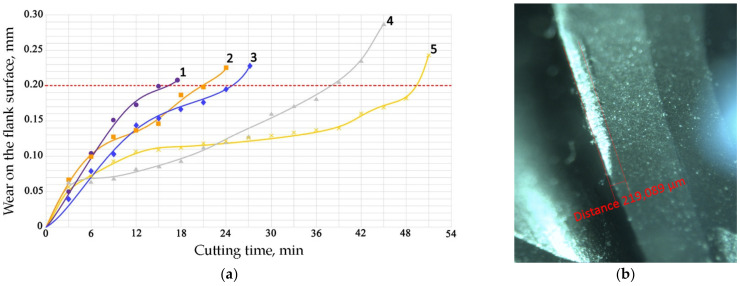
(**a**) The durability of ball-end mills with coatings: 1—uncoated, 2—TiB2, 3—CrN−ncAlCrN/Si_3_N_4_ + DLC, 4—CrN-ncAlCrN/Si_3_N_4_ + TiB2, 5—CrN-ncAlTiN/Si_3_N_4_ + TiB2. The cutting mode is shown in Table 3, the dotted line shows the value of the wear criterion; (**b**) the scheme of flank surface wear measure.

**Figure 4 materials-16-02535-f004:**
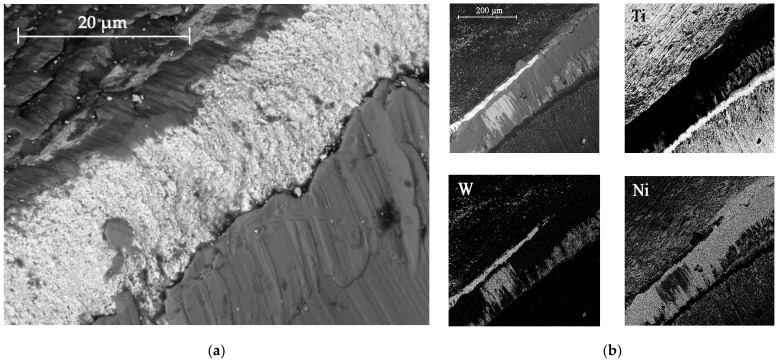
The wear to 0.2 mm on the flank surface cutting edge of the milling cutter tooth coated with TiB_2_ (**a**) and CrN−ncAlCrN/Si_3_N_4_+DLC (**c**) in reflected electrons, (**b**,**d**) distribution of chemical elements on the surface of worn-out milling cutters, respectively.

**Figure 5 materials-16-02535-f005:**
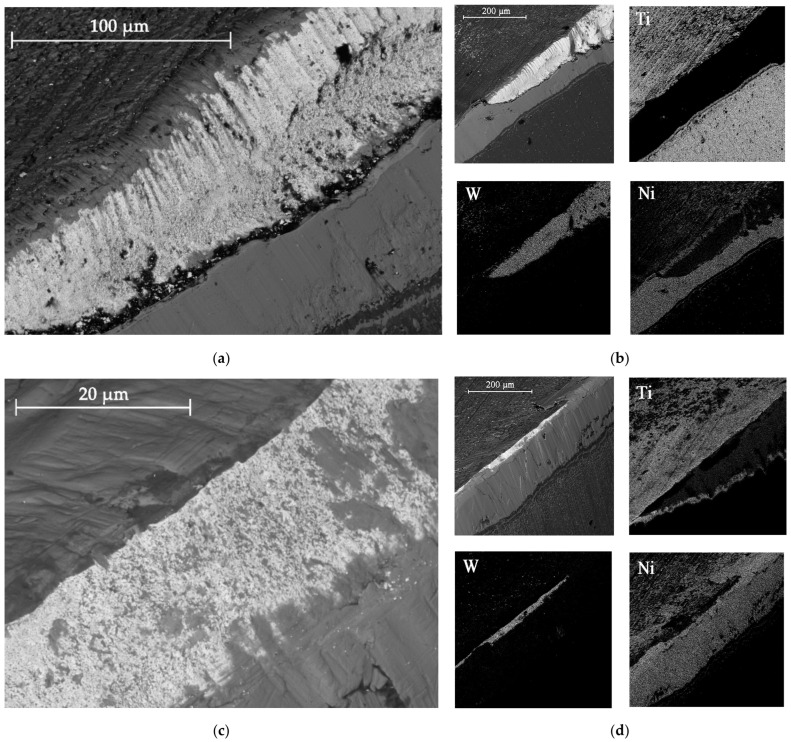
View of the cutting edge worn to 0.2 mm on the flank surface of the milling cutter tooth coated with CrN-ncAlTiN/Si_3_N_4_+Tab2 (**a**) and CrN-ncAlCrN/Si_3_N_4_ + TiB2 (**c**) in reflected electrons and (**b**,**d**) distribution of chemical elements on the surface of worn-out milling cutters, respectively.

**Table 1 materials-16-02535-t001:** The mill’s geometrical parameters.

Parameter	Value
Diameter	11.953 mm
Number of flutes	4
Flute radius	5.992 mm
Primary clearance angle	10.837°
Secondary clearance angle	19.212°
Primary clearance land	0.9528 mm
Rake angle	5.833°

**Table 2 materials-16-02535-t002:** KFM 39 cemented carbide characteristics.

Characteristic	Measuring Unit	Value
Co content	wt%	9.0
HV_30_ microhardness	ISO 3878	1950 ± 50
Crack resistance, K_IC_	MPa·m^1/2^	9.3
Tungsten carbide average particle size	µm	0.4

**Table 3 materials-16-02535-t003:** Cutting mode during the ball and mills test.

Parameter	Value
Cutting speed V_C_	250 m/min
Milling cutter rotation frequency	8657 min^−1^
Feed f_z_	0.05 mm/tooth
Depth of cut a_p_	0.3 mm
The inclination angle of the milling cutter β	40°
Effective milling cutter diameter d_eff_	9.19 mm
Workpiece rotation speed V_W_	1.7 m/min

**Table 4 materials-16-02535-t004:** Parameters of wear-resistant coatings.

Coating	Grinding Image	Surface Condition	Parameters
TiB_2_	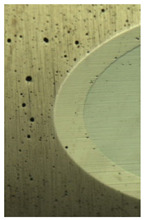	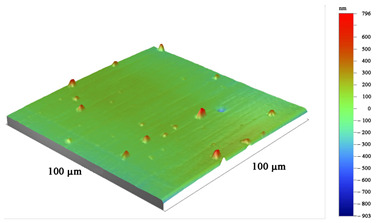	h_TiB2_ = 2.5 µm
			R_A_ = 0.06 µm R_Z_ = 1.15 µm
			HV_10_ = 34.0 GPa
CrN− ncAlTiN/Si_3_N_4_ + TiB_2_	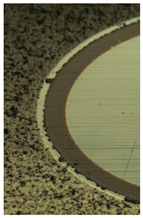	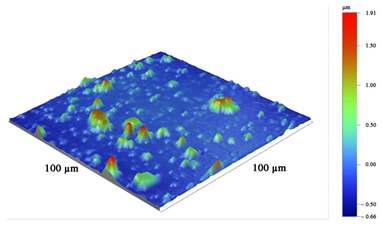	h_TiB2_ = 0.6 µm
			h_ncAlTiN/Si3N4_ = 2.0 µm
			R_A_ = 0.15 µm R_Z_ = 1.69 µm
			HV_10_ = 38.4 GPa
CrN− ncAlCrN/Si_3_N_4_ + TiB_2_	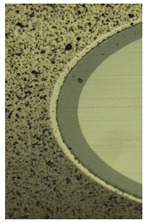	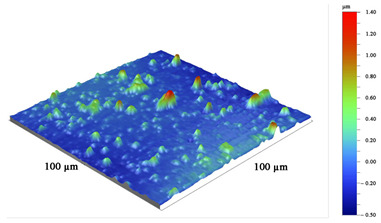	h_TiB2_ = 0.7 µm
			h_ncAlCrN/Si3N4_ = 2.0 µm
			R_A_ = 0.11 µm R_Z_ = 1.62 µm
			HV_10_ = 37.0 GPa
CrN− ncAlCrN/Si_3_N_4_ + DLC	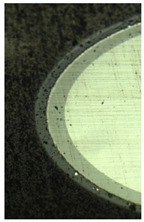	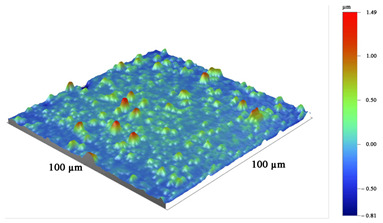	h_DLC_ = 1.0 µm
			h_ncAlCrN/Si3N4_ = 1.7 µm
			R_A_ = 0.13 µm R_Z_ = 2.01 µm
			HV_10_ = 35.6 GPa

**Table 5 materials-16-02535-t005:** Three-dimensional images of cutting edges and their light sections for cutters worn up to 0.2 mm on the back surface with TiB2 and CrN−ncAlCrN/Si_3_N_4_+DLC coatings.

Coating	3D Image	Cross Section
TiB_2_	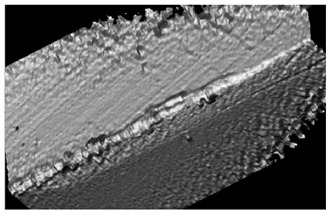	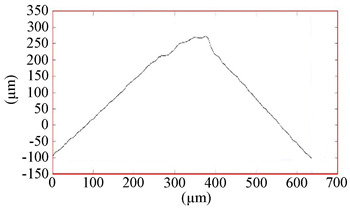
CrN− ncAlCrN/Si_3_N_4_ + DLC	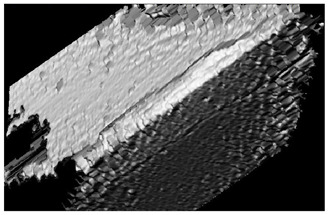	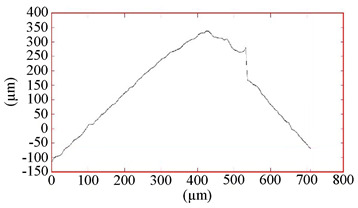

**Table 6 materials-16-02535-t006:** Three-dimensional images of cutting edges and their light cross-section for cutters worn up to 0.2 mm on the flank surface with CrN-ncAlTiN/Si_3_N_4_+TiB2 and CrN-ncAlCrN/Si_3_N_4_+TiB_2_ coatings.

Coating	3D Image	Cross Section
CrN− ncAlTiN/Si_3_N_4_ +TiB_2_	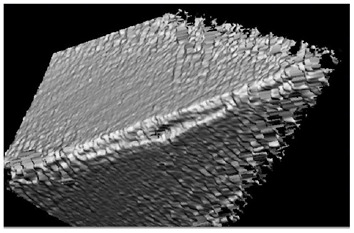	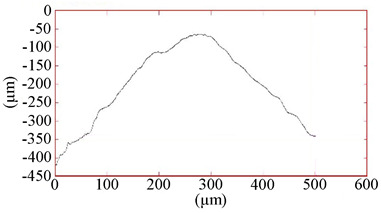
CrN− ncAlCrN/Si_3_N_4_ +TiB_2_	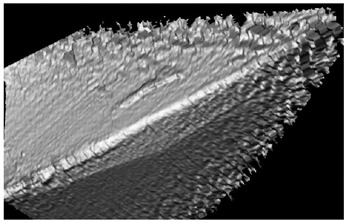	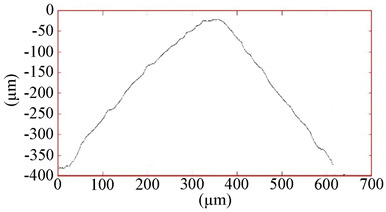

## Data Availability

Not applicable.

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
