# Peer review of "The Study of Radius End Mills with TiB2 Coating When Milling a Nickel Alloy"

_materials, 2023, doi:10.3390/ma16062535_

Round 1

Reviewer 1 Report

1. There are many grammar and spelling errors in the paper, even in the Abstract, such as “which causes accelerated tool wear”, “650 C”, “this coating begins to burn out”, and so on. The English language really should be further improved to increase the paper readability.

2. In Abstract, the author introduced many information about the DLC coating, but the DLC coating is not separately compared in the experiments, this is easy to induce misunderstanding. In my opinion, the Abstract can be revised to more consistent with the main research content of this paper.

3. Four ball end mills have been presented in Figure 1, their different coatings also should be marked in the figure.

4. The ball mill end usually used in milling process, why is the milling-turning experiments carried out in this work, but not the milling experiments? What is the actual relative linear speed between the tool and workpiece?

5. How to measure the flank surface wear of this ball end mill, please provide a image to illustrate.

6. The title of this paper can be revised to be more consistent with the paper content.

Reviewer 2 Report

The authors present a comparative study of a radius end mill covered with antifriction coatings for the milling of nickel alloys. The aim of the study is to compare the radius end mills with antifriction coatings of TiB2 and DLC deposited on a nanocrystalline coating of (Ti(Cr)AlSi)N when milling a heat-resistant nickel alloy. The results are of interest to the processing industry of such alloys but the manuscript has issues that must address by the authors before publishing.

2. Materials and methods

a) It is mentioned that the wear-resistant coatings are presented in Table 3, but the actual Table 3 presents the "Cutting mode during the ball and miles test. It seems that there's a missing Table.

b) A key parameter of the research is the nanocrystalline coating of (Ti(Cr)AlSi)N deposited by a PVD method. Nevertheless, the deposition parameters are not included in this section. Furthermore, the study did not characterize the coatings but the authors justify the nanocrystalline microstructure by citing another study [22].

3. Results and discussion

a) The manuscript includes the superficial and hardness characterization of the TiB2 and DLC films but the elemental and structural composition were not reported. It is suggested to include an XRD and/or Raman characterization to confirm the expected structures of the antifriction coatings.

b) How was determined the thickness of the different coatings? From the images depicted in Table 4?

c) They claim that the growth of the TiB2 layer was 3 times slower in the samples with the (Ti(Cr)AlSi)N coatings than in the hard alloy, based on the electrical and physical properties of the substrate. Nevertheless, in a sputtering process, the deposition rate strongly depends on the deposition parameters than on the properties of the substrate, especially if the coatings were deposited at the same time.

d) The results exhibited in figure 3 show the durability of the different coatings during the cutting tests. The characteristics of the wear surface are shown in figures 4 and 5, from which it is extracted many conclusions like the presence of WC particles or the stick of the nickel alloy on the flank surface. Nevertheless, these conclusions are not clear from the images of the manuscript without the support of EDX analysis, either by punctual or mapping analysis.

e) On Page 9, it seems that there is a mistake in the index of curves 5 and 6.

f) It is not clear which information is extracted from figures 5 and 6, since they are not discussed in the manuscript.

g) For this reviewer, the exhibited results do not support the formation of a tribological film based on boron oxides that would explain the increase of durability of the samples with combined coatings. But, they can be obtained from a deeper SEM analysis of the tool surfaces after the tests.

Round 2

Reviewer 1 Report

All comments have been revised. 

Reviewer 2 Report

The authors have addressed all the reviewer comments and the manuscript is suitable for publication in Materials journal.